# Highly Efficient Inverted Organic Light-Emitting Devices with Li-Doped MgZnO Nanoparticle Electron Injection Layer

**DOI:** 10.3390/mi16060617

**Published:** 2025-05-24

**Authors:** Hwan-Jin Yoo, Go-Eun Kim, Chan-Jun Park, Su-Been Lee, Seo-Young Kim, Dae-Gyu Moon

**Affiliations:** Department of Electronic Materials, Device, and Equipment Engineering, Soonchunhyang University, Asan-si 31538, Republic of Korea; yhjin1209@sch.ac.kr (H.-J.Y.); tkfkdrhdms153@sch.ac.kr (G.-E.K.); cjpark5581@sch.ac.kr (C.-J.P.); 20247114@sch.ac.kr (S.-B.L.); syoung031219@sch.ac.kr (S.-Y.K.)

**Keywords:** OLED, inverted structure, Li-doped MgZnO nanoparticles, electron injection layer

## Abstract

Inverted organic light-emitting devices (OLEDs) have been attracting considerable attention due to their advantages such as high stability, low image sticking, and low operating stress in display applications. To address the charge imbalance that has been known as a critical issue of the inverted OLEDs, Li-doped MgZnO nanoparticles were synthesized as an electron-injection layer of the inverted OLEDs. Hexagonal wurtzite-structured Li-doped MgZnO nanoparticles were synthesized at room temperature via a solution precipitation method using LiCl, magnesium acetate tetrahydrate, zinc acetate dihydrate, and tetramethylammonium hydroxide pentahydrate. The Mg concentration was fixed at 10%, while the Li concentration was varied up to 15%. The average particle size decreased with Li doping, exhibiting the particle sizes of 3.6, 3.0, and 2.7 nm for the MgZnO, 10% and 15% Li-doped MgZnO nanoparticles, respectively. The band gap, conduction band minimum and valence band maximum energy levels, and the visible emission spectrum of the Li-doped MgZnO nanoparticles were investigated. The surface roughness and electrical conduction properties of the Li-doped MgZnO nanoparticle films were also analyzed. The inverted phosphorescent OLEDs with Li-doped MgZnO nanoparticles exhibited higher external quantum efficiency (EQE) due to better charge balance resulting from suppressed electron conduction, compared to the undoped MgZnO nanoparticle devices. The maximum EQE of 21.7% was achieved in the 15% Li-doped MgZnO nanoparticle devices.

## 1. Introduction

Organic light-emitting devices (OLEDs) have revolutionized the display and lighting industries with their excellent characteristics such as high contrast ratio, wide viewing angles, fast response time, and thin thickness [1,2,3,4]. Additionally, OLEDs enable the implementation of transparent, flexible, stretchable, and lightweight displays, making them a highly promising next generation display technology [5,6]. Among OLED technologies, active-matrix OLEDs (AMOLEDs) have been widely used as a key display technology for smartphones, televisions, and monitors due to their low power consumption and high image quality [7,8]. AMOLEDs utilize thin-film transistors (TFTs) to control each pixel, enabling high-speed switching and superior power efficiency compared to passive-matrix OLEDs. Particularly, oxide TFTs are gaining significant attention because they provide high mobility, a simplified manufacturing process, and high device stability [9,10,11]. Since oxide TFTs primarily use n-type oxide semiconductors, there is growing interest in inverted OLEDs, which can mitigate luminance degradation, operational stress, and image sticking issues [12,13,14].

Conventional OLEDs have a structure in which the hole-injecting electrode is located at the bottom and the electron-injecting electrode is positioned at the top of the devices. In contrast, inverted OLEDs adopt an opposite structure, where the cathode is placed at the bottom and the anode at the top. This structure offers advantages such as improved environmental stability, reduced sensitivity to air and moisture, solution processing capability as well as better compatibility with n-type oxide TFT backplanes [15,16,17,18,19]. A major issue in inverted OLEDs is electron injection from the cathode into the organic layer. In inverted OLEDs, indium tin oxide (ITO) is commonly used as a transparent cathode due to its excellent optical transmittance and electrical conductivity [19]. However, ITO has a high work function, making electron injection from the ITO cathode into the organic layer difficult [13,14,15,16,17,18,19]. Therefore, it is important to use a material with efficient electron injection properties as the electron injection layer.

ZnO nanoparticles have been studied to utilize the electron injection layer in the OLEDs because they have high electron mobility, high transparency in visible range, and a low conduction band minimum energy [20,21,22]. Since ZnO nanoparticles are generally dispersed in organic solvents such as methanol and ethanol, solution processes such as spin coating and inkjet printing are typically used to prepare ZnO nanoparticle layer. There have been very few reports on conventional OLEDs using ZnO nanoparticles because the organic solvents in the ZnO solution can damage the underlying organic layers leading to device failure [23,24,25]. In contrast, in an inverted structure, a ZnO nanoparticle layer is formed before coating the organic layers, preventing the organic layers from being damaged by ZnO solution. Therefore, the inverted structure has been predominantly used for ZnO nanoparticles [26,27,28,29,30]. In inverted structure, ZnO nanoparticles are coated onto the ITO cathode, allowing improvements in electron mobility, electrical conductivity, and surface morphology through the processes such as thermal annealing and surface treatment [27,28]. In the fabrication of inverted OLEDs, an additional interlayer such as polyethyleneimine, thin Ag nanoparticle, and Ba(OH)_2_ layer was inserted between the ZnO nanoparticle layer and the electron transport layer to adjust electron injection [31,32,33,34,35]. However, this interlayer increases the complexity of the device fabrication process.

In this paper, we report on inverted OLEDs without an additional interlayer using Li doped MgZnO nanoparticles. Alkali metal-doped ZnO and MgZnO nanoparticles have been widely used in quantum dot light-emitting diodes to control electron injection and transport properties [36,37,38,39]. However, there are very few reports on inverted OLEDs utilizing alkali metal-doped ZnO and MgZnO nanoparticles [40,41]. Manzhi et al. fabricated inverted OLEDs using LiZnO and MgZnO nanoparticles [40,41]. They incorporated LiZnO nanoparticles or MgZnO nanostructures into poly[9,9-dioctylfluorenyl-2,7-diyl] (PFO) to form the nanocomposite emitting layer for the inverted OLEDs. They investigated the current-voltage characteristics of the inverted devices with PFO-LiZnO or PFO-MgZnO nanocomposites. However, they did not investigate the luminance or efficiency characteristics of the inverted devices. In addition, there is no report on the phosphorescent inverted devices with a Li-doped MgZnO nanoparticle layer. In this paper, we used the Li-doped MgZnO nanoparticle layer as an electron injection layer in the inverted phosphorescent devices. Li-doped MgZnO nanoparticles influence electron injection and charge balance in the inverted devices, resulting in highly efficient inverted devices with a maximum external quantum efficiency of 21.7%.

## 2. Materials and Methods

Li-doped MgZnO nanoparticles were synthesized using the solution precipitation method [37]. Magnesium acetate tetrahydrate (≥99%, Sigma Aldrich, St. Louis, MO, USA), lithium chloride (LiCl, ≥99%, Sigma Aldrich), zinc acetate dihydrate (≥99%, Sigma Aldrich), dimethyl sulfoxide (DMSO, ≥99%, Sigma Aldrich), tetramethylammonium hydroxide pentahydrate (TMAH, ≥97%, Sigma Aldrich), and ethyl alcohol were used to synthesize the Li-doped MgZnO nanoparticles. Magnesium acetate tetrahydrate was added to 30 mL of DMSO and stirred at 600 rpm until completely dissolved. LiCl was added to the magnesium acetate tetrahydrate solution and stirred at 600 rpm until completely dissolved. Then, zinc acetate dihydrate was added to the mixed solution of magnesium acetate tetrahydrate and LiCl, and stirred until completely dissolved. An amount of 4.62 mmol of TMAH was completely dissolved in 10 mL of ethanol using a vertex meter. The TMAH solution was added dropwise to the mixed solution of magnesium acetate tetrahydrate, LiCl, and zinc acetate dihydrate with continuous stirring. Then, ethyl acetate was used to precipitate Li-doped MgZnO nanoparticles. After collecting the precipitate using a centrifuge, it was washed repeatedly with ethanol to minimize residual ethyl acetate. Finally, the collected Li-doped MgZnO nanoparticles were redispersed in ethanol. The concentration of Mg in Li-doped MgZnO nanoparticles was fixed at 10%, while the concentration of Li was varied to 10% and 15% by adjusting the amounts of LiCl and zinc acetate dihydrate. For example, 0.15 mmol of LiCl and 1.275 mmol of zinc acetate dihydrate were used to synthesize 10% Li-doped MgZnO nanoparticles. The crystal structure of Li-doped MgZnO nanoparticles was investigated using X-ray diffraction method (Miniflex 600, Rigaku, Tokyo, Japan). Transmission electron microscopy (TEM, JEM-ARM200F, JEOL, Tokyo, Japan) was used to study the particle size and shape of Li-doped MgZnO nanoparticles. The UV-vis absorption spectrum of the nanoparticles was measured using a UV-vis spectrometer (Lamda 35, PerkinElmer, Waltham, MA, USA). The valence band maximum energy of Li-doped MgZnO nanoparticles was measured using ultraviolet photoelectron spectroscopy (UPS, NEXSA, Thermo Fisher Scientific, Waltham, MA, USA). Atomic force microscopy (AFM, XE7, Park Systems, Suwon, Republic of Korea) was utilized to measure the surface roughness of thin films coated with Li-doped MgZnO nanoparticles.

Inverted phosphorescent OLEDs using the synthesized Li-doped MgZnO nanoparticles were fabricated on the ITO-coated glass substrates. The sheet resistance of the ITO film was about 10 Ω/sq. ITO cathode patterns were defined using a standard photolithography process. The ITO cathode patterns were cleaned with acetone, isopropyl alcohol, and deionized water. The cleaned ITO patterns were exposed to oxygen plasma to remove organic residues before spin-coating with Li-doped MgZnO nanoparticles in ethanol. The thickness of the Li-doped MgZnO nanoparticle layer was fixed to be 10 nm. After forming the Li-doped MgZnO nanoparticle layer, organic and metal layers were deposited by the vacuum thermal evaporation method at a base pressure of 10^−6^ Torr. A 25 nm thick 2,2′,2″-(1,3,5-benzinetriyl)-tris-(1-phenyl)-1-H-benzimidazole (TPBi) electron transport layer was deposited, followed by a 20 nm thick 4,4′-bis(9-carbazolyl)biphenyl (CBP) host layer co-deposited with a green phosphorescent dopant, tris(2-phenylpyridine)iridium(III) [Ir(ppy)_3_], for the emission layer. The concentration of Ir(ppy)_3_ was 5 wt%. After depositing the CBP:Ir(ppy)_3_ emission layer, a 30 nm-thick undoped CBP hole transport layer was deposited, followed by a 10 nm-thick MoO_3_ hole injection layer was deposited. Finally, a 100 nm thick Al layer was evaporated to define the anode through a shadow mask without breaking the vacuum. The completed device structure is ITO/Li-doped MgZnO (10 nm)/TPBi (25 nm)/CBP:Ir(ppy)_3_ (20 nm, 5 wt%)/CBP (30 nm)/MoO_3_ (10 nm)/Al. Current density-voltage-luminance (J-V-L) characteristics of the inverted devices were measured using computer-controlled Keithley 2400 source-measure units and a calibrated Si photodiode. The electroluminescence spectra of the device were measured using a spectroradiometer (CS1000, Minolta, Tokyo, Japan).

## 3. Results and Discussion

Li-doped MgZnO nanoparticles were prepared by solution precipitation method using LiCl, magnesium acetate tetrahydrate, and TMAH for the electron injection layer in inverted OLEDs. The concentration of Mg was fixed at 10%, and the concentration of Li was varied to be 10% and 15% in the Li-doped MgZnO nanoparticles. Figure 1 shows the X-ray diffraction patterns for the synthesized Li-doped MgZnO nanoparticles. The undoped MgZnO nanoparticles exhibited diffraction peaks at 2θ values of 31.6°, 34.2°, 36.0°, 47.3°, 56.4°, 62.5°, and 67.8°, which correspond to the (100), (002), (101), (102), (110), (103), and (112) planes of the hexagonal wurtzite structure of ZnO (JCPDS card 361451). The X-ray diffraction pattern of the MgZnO indicates that doping with 10% Mg into ZnO nanoparticles does not alter the hexagonal wurtzite structure of ZnO [42,43,44]. Even with 10% and 15% Li doping, diffraction peaks corresponding to the (100), (101), (102), (110), (103), and (112) planes of the hexagonal wurtzite structure of ZnO are observed at diffraction angles (2θ) of 31.6°, 36.1°, 47.3°, 56.4°, 62.4°, 67.9°, respectively. As shown in the figure, the X-ray diffraction pattern remains unchanged with Li doping into MgZnO, while the diffraction peaks become broader. This result indicates that the hexagonal wurtzite structure of MgZnO is maintained with Li doping, whereas the particle size of Li-doped MgZnO nanoparticles is smaller than that of the undoped MgZnO nanoparticles [45]. 

Figure 2 shows the TEM images and the particle size distributions of the synthesized Li-doped MgZnO nanoparticles. MgZnO nanoparticles have an elliptical shape, and the Li-doped MgZnO nanoparticles also retain this elliptical morphology, indicating that Li-doping into the MgZnO nanoparticles does not cause significant changes in particle shape. The average particle size of MgZnO nanoparticles was 3.6 nm, while it decreased to 3.0 nm for 10% Li-doped MgZnO and to 2.7 nm for 15% Li-doped MgZnO nanoparticles, indicating that the particle size decreases with increasing Li doping. The particle size distribution with Li doping also shows a similar trend to that of the particle size. MgZnO nanoparticles show the highest count in the particle size range of 3.5 to 4.0 nm. With Li doping, the dominant count shifts toward smaller particle sizes. The 10% Li-doped MgZnO nanoparticles show the highest count in the particle size range of 2.8 to 3.3 nm, while the 15% Li-doped MgZnO nanoparticles exhibit the highest count in the range of 2.3 to 2.8 nm. The Li-doped MgZnO nanoparticles also exhibit a more uniform particle size compared to the MgZnO nanoparticles. MgZnO nanoparticles have particle sizes ranging from 2.0 to 5.5 nm, whereas the 10% Li-doped MgZnO nanoparticles exhibit particle sizes between 1.8 and 4.8 nm, and the 15% Li-doped MgZnO nanoparticles have particle sizes in the range of 1.8 to 4.3 nm. For the synthesis of MgZnO nanoparticles, magnesium acetate tetrahydrate, zinc acetate dihydrate, and TMAH were used as the Mg and Zn precursors and the oxidizing agent, respectively. Mg^2+^ and Zn^2+^ ions from magnesium acetate tetrahydrate and zinc acetate dihydrate react with OH^−^ ions from TMAH to form MgZnO nanoparticles. The MgZnO nanoparticles grow through the Ostwald ripening process [46,47]. For the synthesis of Li-doped MgZnO nanoparticles, LiCl was used as the Li precursor. It is assumed that Li^+^ ions from LiCl are adsorbed onto the surface of the MgZnO particles, thereby interfering with the recombination of Mg^2+^ and Zn^2+^, and OH^−^ ions. As a result, the growth rate of the nanoparticles decreases, leading to a reduction in particle size. From Figure 1 and Figure 2, it can be observed that Li doping in MgZnO nanoparticles does not alter the crystal structure, whereas the particle size decreases. MgZnO is thermodynamically stable with a hexagonal wurtzite structure. The ionic radius of Zn^2+^ is approximately 0.74 Å, while that of Li^+^ is 0.59 Å [48]. Li^+^ ions can either substitute for Zn^2+^ ions or occupy interstitial sites, causing only minimal lattice distortion within the solubility limit. Therefore, the crystal structure remains unchanged by Li doping [41,49]. On the other hand, Li^+^ ions readily adsorb onto the surface of MgZnO nanoparticles during crystal growth. These adsorbed Li^+^ ions hinder the approach of Zn^2+^ ions to the nanoparticle surface and suppress the reaction between Zn^2+^ and OH^−^ ions, thereby reducing the crystal growth rate [46,47]. As a result, the particle size decreases.

Figure 3 shows the UV absorption spectra of the Li-doped MgZnO nanoparticles. As the Li concentration increases, the absorption spectrum shifts toward shorter wavelengths. While the absorption edge of MgZnO is at 347 nm, those of the 10% and 15% Li-doped MgZnO nanoparticles are at 339 and 337 nm, respectively. The blue shift of the absorption spectrum indicates an increase in the band gap. The band gap energy was determined from the intersection point between the tangent to the linear section of the absorption spectrum and the wavelength axis [20]. The measured band gaps of MgZnO, 10% and 15% Li-doped MgZnO nanoparticles are 3.57, 3.66, and 3.68 eV, respectively. This increase in the band gap is believed to be related to the reduction in particle size caused by Li-doping as shown in Figure 2. Li-doped MgZnO nanoparticles with sizes ranging from 2.7 to 3.6 nm exhibit quantum confinement effects, leading to an increase in band gap as the particle size decreases [50,51,52,53]. In addition, an increase in the band gap has been observed when Li is doped into bulk ZnO [54,55]. Similarly, it can be suggested that Li doping into MgZnO nanoparticles appears not only to increase the band gap due to a reduction in particle size but also contribute to an additional increase in band gap caused by the Li doping itself.

Figure 4 shows the UPS spectra corresponding to the secondary cut-off and the valence band edge region for MgZnO, 10% Li-doped MgZnO, and 15% Li-doped MgZnO nanoparticles. The valence band maximum (VBM) was determined using the equation VBM = 21.22 − (E_cutoff_ − E_onset_), where 21.22 eV is the incident photon energy, E_cutoff_ is the high binding energy cutoff, and E_onset_ is the onset energy of the valence band region [56]. The VBM energies for MgZnO, 10% Li-doped MgZnO, and 15% Li-doped MgZnO are 6.49, 6.59, and 6.69 eV, respectively. The VBM energy increases with Li doping. The position of the conduction band minimum (CBM) was calculated using the equation, E_c_ = E_v_ − E_g_, where E_c_ is the CBM energy, E_v_ is the VBM energy obtained from UPS spectra, and E_g_ is the band gap determined from the UV-vis absorption spectrum. The CBM energies for MgZnO, 10% Li-doped, and 15% Li-doped MgZnO are 2.92, 2.93, and 3.01 eV, respectively, showing a slight increase with Li doping. The increase in CBM energy indicates a higher energy barrier for electron injection from the Li-doped MgZnO to the organic electron transport layer.

Figure 5 shows the photoluminescence (PL) spectra of MgZnO, 10% and 15% Li-doped MgZnO nanoparticles. As shown in Figure 3, since the band gap of Li-doped MgZnO nanoparticles exceeds 3.5 eV, the visible emission is presumed to originate mainly from surface defects [57]. MgZnO nanoparticles exhibit a broad emission peak around 523 nm. With Li doping, the emission peak shifts toward the shorter wavelength region, appearing at 507 nm and 494 nm for 10% and 15% Li-doped MgZnO nanoparticles, respectively. The visible emission wavelength of MgZnO nanoparticles depends on their band gap [20,58]. As shown in Figure 3, the band gap increases with Li doping, resulting in a blue shift of the defect related emission wavelength. This occurs because the increased band gap shifts the conduction and valence band edges while the absolute position of the defect level remains constant, leading to a higher energy difference between defect level and the conduction band minimum, as reported by Pan et al. [20]. In addition, the PL intensity decreases with Li doping. Since PL intensity depends on the amount of surface defects, the lower PL intensity in Li-doped MgZnO nanoparticles indicates the reduced surface defects [30,57]. During the formation of nanoparticles, the Li ions may be attached to the surface and passivate the surface defects. A similar phenomenon has also been observed when Mg is doped into ZnO nanoparticles [44].

Figure 6 shows the AFM images of the thin films coated with MgZnO, 10% and 15% Li-doped MgZnO nanoparticles. The root mean square (RMS) surface roughness of the MgZnO nanoparticle film is 2.6 nm, while the 10% and 15% Li-doped MgZnO films exhibit RMS surface roughness of 2.4 nm and 2.8 nm, respectively. On the other hand, the particle sizes shown in Figure 2 are 3.6, 3.0, and 2.7 nm for the undoped, 10% and 15% Li-doped MgZnO nanoparticles, resulting in the discrepancy between the particle size observed by TEM and the surface roughness measured by AFM. This apparent inconsistency can be attributed to the agglomeration of nanoparticles [59]. As the particle size decreases, the surface energy of nanoparticles increases significantly, which promotes agglomeration during film formation. Especially for the smallest particles of 2.7 nm, the reduced stability leads to local agglomeration, resulting in uneven surface morphology and increased surface roughness, despite the smaller individual particle size.

To investigate the electrical conduction properties of Li-doped MgZnO nanoparticles, ITO/Li-doped MgZnO nanoparticles (80 nm)/Al devices were prepared. Figure 7 shows the current density–voltage curves of the devices. Compared to undoped MgZnO nanoparticles, the Li-doped MgZnO nanoparticles exhibit reduced current transport. For example, MgZnO nanoparticle device shows a current density of 544 mA/cm^2^ at 5 V, whereas the 10% and 15% Li-doped MgZnO nanoparticles exhibit current densities of 321 and 79 mA/cm^2^, respectively, at the same voltage. The electron mobility of the Li-doped MgZnO nanoparticles was estimated in the space charge limited current region using the equation, J = (9/8)με_o_ε_r_(V^2^/L^3^), where J is the current density, μ is the electron mobility, ε_o_ is the dielectric constant of free space, ε_r_ is the relative dielectric constant of the nanoparticle film, V is the applied voltage, and L is the thickness of the nanoparticle film [60]. The mobility of the undoped MgZnO nanoparticle device was approximately 10^−5^ cm^2^/Vs, while the mobilities of the 10% and 15% Li-doped MgZnO nanoparticle devices were 5 × 10^−6^ and 9 × 10^−7^ cm^2^/Vs, respectively, indicating a decrease in mobility with increasing Li concentration. This reduction in mobility is presumed to be related to the decreased particle size and impurity effects induced by Li doping. As the particle size decreases with Li doping as shown in Figure 2, the electron scattering at the particle surfaces increases. Additionally, impurity scattering due to Li dopants also contributes to the decrease in mobility.

Inverted OLEDs were fabricated using Li-doped MgZnO nanoparticles. The device structure is ITO/Li-doped MgZnO nanoparticles (10 nm)/TPBi (25 nm)/CBP:Ir(ppy)_3_ (20 nm, 5 wt%)/CBP (30 nm)/MoO_3_ (10 nm)/Al. Figure 8 shows the device structure and energy diagram of the inverted devices with Li-doped MgZnO nanoparticles. Figure 9 shows the electroluminescence spectra measured at various voltages for the inverted devices fabricated using MgZnO, 10% and 15% Li-doped MgZnO nanoparticles. The devices exhibit a strong emission at 515 nm, which originates from the triplet excited state of Ir(ppy)_3_ [61]. The emission from Ir(ppy)_3_ remains unchanged with varying voltage. In addition, a weak emission peak can be observed around 400–410 nm. This peak corresponds to the emission from CBP and increases with increasing voltage [62]. The CBP emission peak indicates electron leakage from the emission layer into the CBP hole transport layer, implying that charge balance is dominated by electron conduction. Furthermore, this result suggests that the recombination zone is located near the emission layer adjacent to the CBP hole transport layer. As the voltage increases, the CBP emission peak becomes more pronounced, indicating increased electron leakage into CBP layer. Moreover, the devices based on 10% and 15% Li-doped MgZnO nanoparticles exhibit weaker CBP emission intensity compared to the device with undoped MgZnO nanoparticles. This suggests that charge balance is improved in the devices using Li-doped MgZnO nanoparticles.

Figure 10 shows the current density–voltage–luminance and external quantum efficiency (EQE) curves of the inverted OLEDs using Li-doped MgZnO nanoparticles. It can be observed that the driving voltage of the Li-doped MgZnO nanoparticle devices is higher. For example, in the case of the undoped MgZnO device, the driving voltage for a current density of 20 mA/cm^2^ is 12.4 V. In contrast, the devices with 10% and 15% Li-doped ZnO nanoparticles have driving voltages of 13.8 V and 14.4 V, respectively, to achieve the same current density. This increase in driving voltage is consistent with the suppression of electron transport in Li-doped MgZnO nanoparticle, as shown in Figure 7. The turn-on voltage for obtaining a luminance of 1 cd/m^2^ also slightly increases with Li doping. The turn-on voltages for MgZnO, 10%, and 15% Li-doped MgZnO nanoparticle devices are 6.4, 6.8, and 6.8 V, respectively. The turn-on voltages are quite high compared to the other inverted OLEDs [13,14,15,16,63]. For example, Hwang et al. reported turn-on voltages of 3.4–3.8 V in the inverted devices with ZnO nanoparticles [63]. The high turn-on voltages in the Li-doped MgZnO devices may be attributed to the low electron mobilities of both the Li-doped MgZnO nanoparticles and the TPBi electron transport layer. The electron mobility of ZnO nanoparticles has been reported to be approximately 6 × 10^−4^ to 5 × 10^−3^ cm^2^/Vs [20]. On the other hand, the electron mobility of Li-doped MgZnO nanoparticles is estimated to be in the range of 9 × 10^−7^ to 10^−5^ cm^2^/Vs, based on Figure 7. In addition, the electron mobility of TPBi ranges from 5.6 × 10^−8^ to 2.1 × 10^−5^ cm^2^/Vs [64]. These low mobilities lead to a significant voltage drop across the Li-doped MgZnO and TPBi layers, resulting in high turn-on voltages. The voltage required for high luminance also increases with Li doping. For example, in the case of MgZnO, the voltage for 1000 cd/m^2^ luminance is 9.6 V, whereas for 10% and 15% Li-doped MgZnO devices, it increases to 11.8 V and 12.4 V, respectively. Figure 10b shows the EQE curves for the Li-doped MgZnO nanoparticle devices. The maximum EQE of the Li-doped MgZnO devices is higher than that of the undoped devices. The MgZnO device shows a maximum EQE of 17.5% at 296 cd/m^2^, while the 10% and 15% Li-doped MgZnO nanoparticle devices show maximum EQEs of 18.5% at 313 cd/m^2^ and 21.7% at 683 cd/m^2^, respectively, indicating better charge balance in the Li-doped MgZnO nanoparticle devices.

## 4. Conclusions

Li-doped MgZnO nanoparticles with a hexagonal wurtzite structure were synthesized using LiCl, magnesium acetate tetrahydrate, zinc acetate dihydrate, and TMAH for the electron injection layer in the inverted OLEDs. Li doping inhibits particle growth, resulting in smaller particle sizes. The average particle size of MgZnO nanoparticles was 3.6 nm, whereas those of 10% and 15% Li-doped MgZnO nanoparticles were 3.0 and 2.7 nm, respectively. The band gap of Li-doped MgZnO nanoparticles was 3.57 eV, whereas the band gap increased due to the enhanced quantum confinement effects resulting from the reduced particle size, reaching 3.68 eV for the 15% Li-doped ZnO nanoparticles. Li doping also raised both the VBM and CBM energy levels. The VBM energy of MgZnO nanoparticles was 6.49 eV, while those of 10% and 15% Li-doped MgZnO nanoparticles were 6.59 and 6.69 eV, respectively. The CBM energy of MgZnO nanoparticles was 2.92 eV and increased to 3.01 eV with 15% Li doping. The increase in CBM energy due to Li doping indicates an increase in energy barrier for the electron injection from the nanoparticle layer into the TPBi electron transport layer. The Li-doped MgZnO nanoparticles exhibited a visible emission due to the surface defects, and the emission spectrum shifted toward shorter wavelength region with Li doping, accompanied by a decrease in emission intensity. The electron mobility of MgZnO nanoparticles was about 10^−5^ cm^2^/Vs, which decreased with Li doping to 5 × 10^−6^ and 9 × 10^−7^ cm^2^/Vs for the 10% and 15% Li-doped MgZnO nanoparticles, respectively. The inverted phosphorescent OLEDs using Li-doped MgZnO nanoparticles exhibited a suppressed electron conduction, resulting in better charge balance and higher external quantum efficiency compared to the devices using undoped MgZnO nanoparticles. The maximum EQE reached 21.7% for the device with 15% Li-doped MgZnO nanoparticles.

## Figures and Tables

**Figure 1 micromachines-16-00617-f001:**
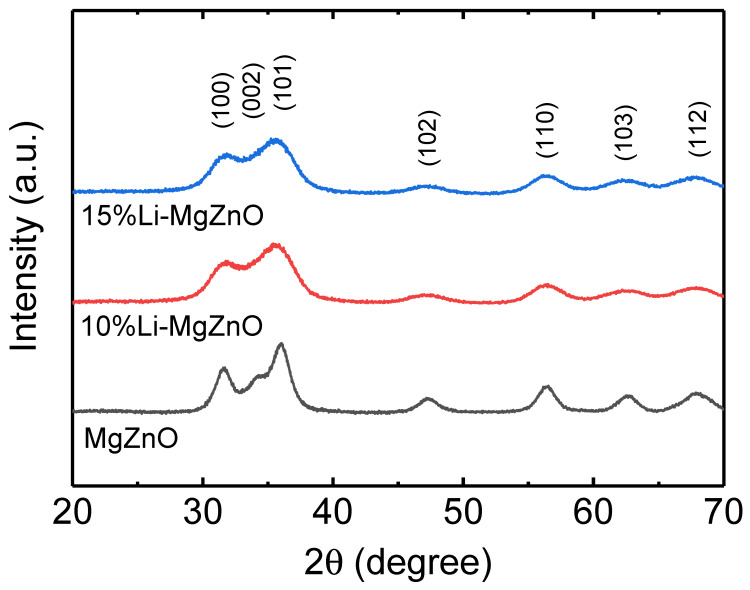
X-ray diffraction patterns of the Li-doped MgZnO nanoparticles.

**Figure 2 micromachines-16-00617-f002:**
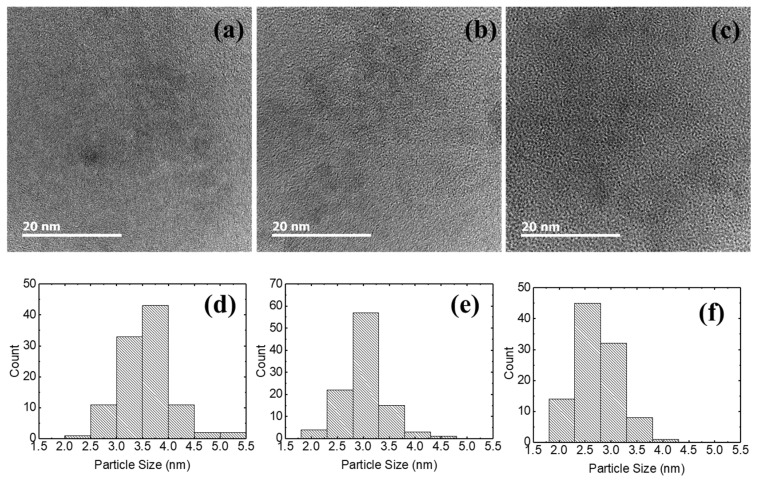
TEM images and particle size distributions of (**a**,**d**) MgZnO nanoparticles, (**b**,**e**) 10% Li-doped MgZnO nanoparticles, and (**c**,**f**) 15% Li-doped MgZnO nanoparticles.

**Figure 3 micromachines-16-00617-f003:**
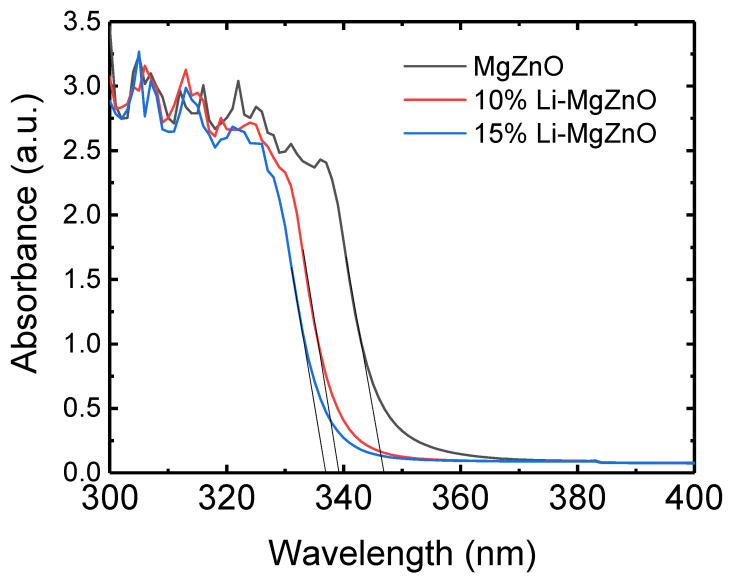
UV-vis absorption spectra of the Li-doped MgZnO nanoparticles.

**Figure 4 micromachines-16-00617-f004:**
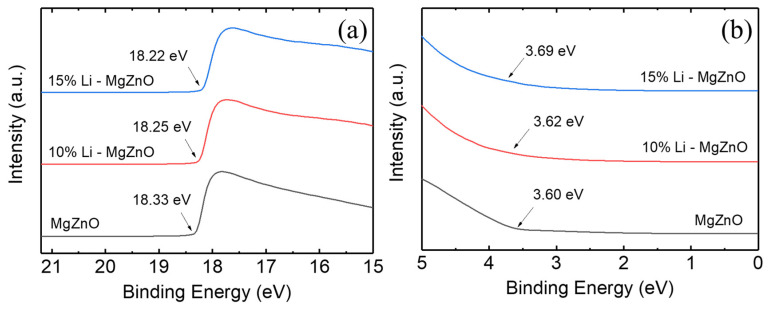
UPS spectra of (**a**) the secondary cutoff region and (**b**) the valence band maximum region for the Li-doped MgZnO nanoparticles.

**Figure 5 micromachines-16-00617-f005:**
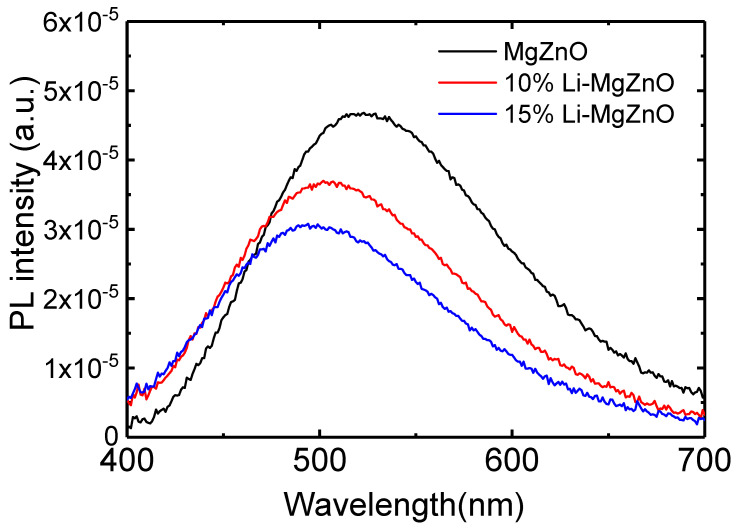
PL spectra of the Li-doped ZnO nanoparticles.

**Figure 6 micromachines-16-00617-f006:**
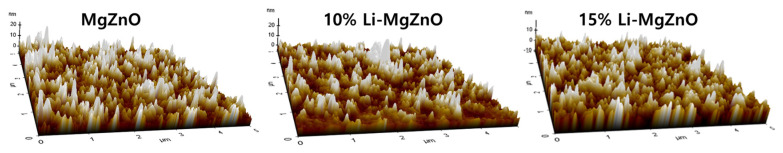
AFM images of MgZnO, 10% Li-doped MgZnO, and 15% Li-doped MgZnO nanoparticle films.

**Figure 7 micromachines-16-00617-f007:**
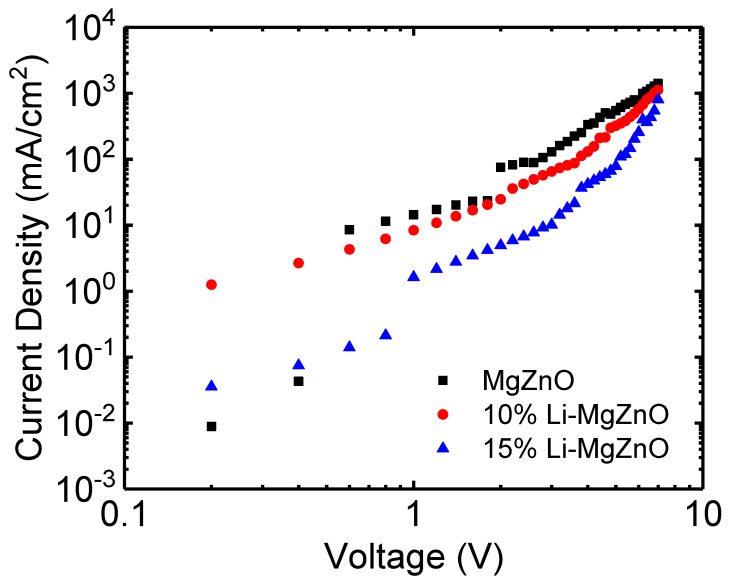
Current density–voltage curves of the Li-doped MgZnO nanoparticle films.

**Figure 8 micromachines-16-00617-f008:**
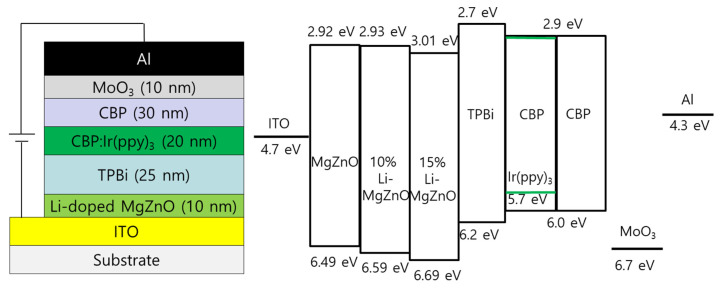
Device structure and energy diagram of the inverted OLEDs with Li-doped MgZnO nanoparticles.

**Figure 9 micromachines-16-00617-f009:**
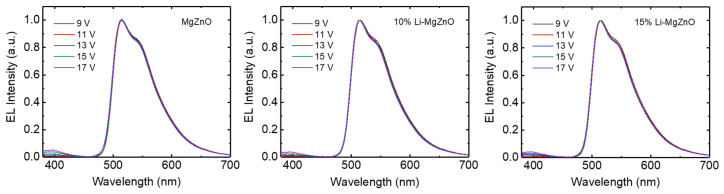
EL spectra at various voltages for the inverted OLEDs with Li-doped MgZnO nanoparticles.

**Figure 10 micromachines-16-00617-f010:**
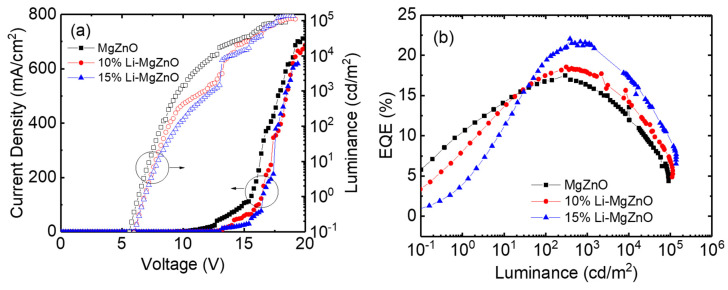
(**a**) Current density—voltage—luminance and (**b**) EQE curves for the inverted OLEDs with Li-doped MgZnO nanoparticles.

## Data Availability

Data are contained within the article.

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
