# Peer review of "Highly Efficient Inverted Organic Light-Emitting Devices with Li-Doped MgZnO Nanoparticle Electron Injection Layer"

_micromachines, 2025, doi:10.3390/mi16060617_

Round 1

Reviewer 1 Report

Comments and Suggestions for Authors

Figure 1 and Figure 2

XRD revealed uncharge of hexagonal structures, while particle size was reduced. Why? Please explain?

Line 217~218

CBM calculation need explained in detail or citation.

Figure 5

PL wavelength revealed blue shift due to increasing band gap energy. However, the PL intensity showed decrease with increasing doping concentration. Why? Please explain.

Figure 6

TEM show reducing particle size of 3.6nm, 3.0nm and 2.7nm, however, the AFM revealed roughness of 2.6, 2.4 and 2.8. Please explain.

Line 255~256

The calculation of mobility need explained in detail or citation.

Author Response

Response to Reviewer 1 Comments:

Thank you very much for taking the time to review this manuscript. We revised the manuscript as reviewer’s comments and suggestions. Please find the detailed responses below and the corresponding revisions in the re-submitted files.

Reviewer #1:

Comments 1: (Figure 1 and Figure 2) XRD revealed unchange of hexagonal structures, while particle size was reduced. Why? Please explain?

Response 1: Thank you for pointing this out. According to reviewer’s comments, we added the sentences and references in the Results and Discussion section as follows: From Figure 1 and Figure 2, it can be observed that Li doping in MgZnO nanoparticles does not alter the crystal structure, whereas the particle size decreases. MgZnO is thermodynamically stable with a hexagonal wurtzite structure. The ionic radius of Zn²⁺ is approximately 0.74 Å, while that of Li⁺ is 0.59 Å [48]. Li⁺ ions can either substitute for Zn²⁺ ions or occupy interstitial sites, causing only minimal lattice distortion within the solubility limit. Therefore, the crystal structure remains unchanged by Li doping [41,49]. On the other hand, Li⁺ ions readily adsorb onto the surface of MgZnO nanoparticles during crystal growth. These adsorbed Li⁺ ions hinder the approach of Zn²⁺ ions to the nanoparticle surface and suppress the reaction between Zn²⁺ and OH⁻ ions, thereby reducing the crystal growth rate [46,47]. As a result, the particle size decreases.

Comments 2: (Line 217~218) CBM calculation need explained in detail or citation.

Response 2: Thank you for comments. According to reviewer’s suggestions, we revised the sentences in the Results and Discussion section as follows: The position of the conduction band minimum (CBM) was calculated using the equation, Ec = Ev – Eg, where Ec is the CBM energy, Ev is the VBM energy obtained from UPS spectra, and Eg is the band gap determined from the UV-vis absorption spectrum.

Comments 3: (Figure 5) PL wavelength revealed blue shift due to increasing band gap energy. However, the PL intensity showed decrease with increasing doping concentration. Why? Please explain.

Response 3: We sincerely appreciate the reviewer’s highly detailed and insightful comments. According to reviewer’s comments, we revised the sentences in the Results and Discussion section as follows: As shown in Figure 3, the band gap increases with Li doping, resulting in blue shift of the defect related emission wavelength. This occurs because the increased band gap shifts the conduction and valence band edges while the absolute position of the defect level remains constant, leading to a higher energy difference between defect level and the conduction band minimum, as reported by Pan et. al [20]. In addition, the PL intensity decreases with Li doping. Since PL intensity depends on the amount of surface defects, the lower PL intensity in Li-doped MgZnO nanoparticles indicates the reduced surface defects. During the formation of nanoparticles, the Li ions may be attached to the surface and passivate the surface defects. A similar phenomenon has also been observed when Mg is doped into ZnO nanoparticles [44].

Comments 4: (Figure 6) TEM shows reducing particle size of 3.6 nm, 3.0 nm and 2.7 nm, however, the AFM revealed roughness of 2.6, 2.4 and 2.8. Please explain.

Response 4: We appreciate the reviewer’s insightful comments regarding the discrepancy between the particle size observed by TEM and the surface roughness measured by AFM. According to reviewer’s comments, we revised the sentences in the Results and Discussion section as follows: The root mean square (RMS) surface roughness of the MgZnO nanoparticle film is 2.6 nm, while the 10% and 15% Li-doped MgZnO films exhibit RMS surface roughness of 2.4 nm and 2.8 nm, respectively. On the other hand, the particle sizes shown in Figure 2 are 3.6, 3.0, and 2.7 nm for the undoped, 10% and 15% Li-doped MgZnO nanoparticles, resulting in the discrepancy between the particle size observed by TEM and the surface roughness measured by AFM. This apparent inconsistency can be attributed to the agglomeration of nanoparticles [59]. As the particle size decreases, the surface energy of nanoparticles increases significantly, which promotes agglomeration during film formation. Especially for the smallest particles of 2.7 nm, the reduced stability leads to local agglomeration, resulting in uneven surface morphology and increased surface roughness, despite the smaller individual particle size.

Comments 5. (Line 255~256) The calculation of mobility needs explained in detail or citation.

Response 5: Thank you for pointing this out. According to reviewer’s comments, we revised the sentences in the Results and Discussion section as follows: The electron mobility of the Li-doped MgZnO nanoparticles was estimated from the space charge limited current equation, J = (9/8)meoer(V2/L3), where J is the current density, m is the electron mobility, eo is the dielectric constant of free space, er is the relative dielectric constant of the nanoparticle film, V is the applied voltage, and L is the thickness of the nanoparticle film [60].

The manuscript has been substantially revised by taking care of all issues raised by the reviewer.

Reviewer 2 Report

Comments and Suggestions for Authors

Authors reported an inverted organic light emitting diode (OLED) with Li-doped MgZnO nanoparticles (NP) as an electron injection layer. There are many reports for an inverted structure in the quantum dot light emitting diode (QLED) using ZnO and MgZnO NP. However, the reports about inverted OLED are less than them. The synthesis and characterization of Li-doped MgZnO are good. I recommend this manuscript to micromachines with a minor revision.

  1. What is the three lines in Figure 7? There is a possibility of readers’ misreading as SCLC region. For red spots from 0.2V to 2V, the real data is proportional to the applied voltage like the ohmic contact region.
  2. Line 297: The definition of turn-on voltage is missing. And overall, the turn-on voltages for the inverted OLEDs in this work are quite high. The discussion and comparison with other inverted OLEDs are needed.

Author Response

Response to Reviewer 2 Comments:

Thank you very much for taking the time to review this manuscript. We revised the manuscript as reviewer’s comments and suggestions. Please find the detailed responses below and the corresponding revisions in the re-submitted files.

Reviewer #2: Authors reported an inverted organic light emitting diode (OLED) with Li-doped MgZnO nanoparticles (NP) as an electron injection layer. There are many reports for an inverted structure in the quantum dot light emitting diode (QLED) using ZnO and MgZnO NP. However, the reports about inverted OLED are less than them. The synthesis and characterization of Li-doped MgZnO are good. I recommend this manuscript to micromachines with a minor revision.

Comments 1: What is the three lines in Figure 7? There is a possibility of readers’ misreading as SCLC region. For red spots from 0.2V to 2V, the real data is proportional to the applied voltage like the ohmic contact region.

Response 1: Thank you for pointing this out. According to reviewer’s comments, we deleted the three lines in Figure 7.

Comments 2: Line 297: The definition of turn-on voltage is missing. And overall, the turn-on voltages for the inverted OLEDs in this work are quite high. The discussion and comparison with other inverted OLEDs are needed.

Response 2: We sincerely appreciate the reviewer’s highly detailed and insightful comments. According to reviewer’s comments, we added the definition of turn-on voltage as the voltage for a luminance of 1 cd/m2. As reviewer points out, the turn-on voltages for the inverted OLEDs in this work are quite high compared to the other inverted ones. We added a discussion and comparison with other inverted OLEDs in the Results and Discussion section as follows: The turn-on voltage for obtaining a luminance of 1 cd/m2 also slightly increases with Li doping. The turn-on voltages for MgZnO, 10%, and 15% Li-doped MgZnO nanoparticle devices are 6.4, 6.8, and 6.8 V, respectively. The turn-on voltages are quite high compared to the other inverted OLEDs [13-16, 63]. For example, Hwang et al. reported turn-on voltages of 3.4 – 3.8 V in the inverted devices with ZnO nanoparticles [63]. The high turn-on voltages in the Li-doped MgZnO devices may be attributed to the low electron mobilities of both the Li-doped MgZnO nanoparticles and the TPBi electron transport layer. The electron mobility of ZnO nanoparticles has been reported to be approximately 6 ´ 10-4 to 5 ´ 10-3 cm2/Vs [20]. On the other hand, the electron mobility of Li-doped MgZnO nanoparticles is estimated to be in the range of 9 ´ 10-7 to 10-5 cm2/Vs, based on Figure 7. In addition, the electron mobility of TPBi ranges from 5.6 ´ 10-8 to 2.l ´ 10-5 cm2/Vs [64]. These low mobilities lead to a significant voltage drop across the Li-doped MgZnO and TPBi layers, resulting in high turn-on voltages.

The manuscript has been substantially revised by taking care of all issues raised by the reviewer.

Round 2

Reviewer 1 Report

Comments and Suggestions for Authors

Good to publish.